# Dynamic Co-Evolution of Cancer Cells and Cancer-Associated Fibroblasts: Role in Right- and Left-Sided Colon Cancer Progression and Its Clinical Relevance

**DOI:** 10.3390/biology11071014

**Published:** 2022-07-06

**Authors:** Sahira Syamimi Ahmad Zawawi, Marahaini Musa

**Affiliations:** Human Genome Centre, School of Medical Sciences, Universiti Sains Malaysia, Kota Bharu 16150, Malaysia; sahirasyamimi002@gmail.com

**Keywords:** colon cancer, activated fibroblast, evolution, heterogeneity, sidedness

## Abstract

**Simple Summary:**

The versatile crosstalk between cancer cells and cancer-associated fibroblasts (CAFs) of the tumour microenvironment (TME) drives colorectal carcinogenesis and heterogeneity. Colorectal cancer (CRC) can be classified by the anatomical sites from which the cancer arises, either from the right or left colon. Although the cancer cell–CAF interaction is being widely studied, its role in the progression of cancer in the right and left colon and cancer heterogeneity are still yet to be elucidated. Further insight into the complex interaction between different cellular components in the cancer niche, their evolutionary process and their influence on cancer progression would propel the discovery of effective targeted CRC therapy.

**Abstract:**

Cancer is a result of a dynamic evolutionary process. It is composed of cancer cells and the tumour microenvironment (TME). One of the major cellular constituents of TME, cancer-associated fibroblasts (CAFs) are known to interact with cancer cells and promote colorectal carcinogenesis. The accumulation of these activated fibroblasts is linked to poor diagnosis in colorectal cancer (CRC) patients and recurrence of the disease. However, the interplay between cancer cells and CAFs is yet to be described, especially in relation to the sidedness of colorectal carcinogenesis. CRC, which is the third most commonly diagnosed cancer globally, can be classified according to the anatomical region from which they originate: left-sided (LCRC) and right-sided CRC (RCR). Both cancers differ in many aspects, including in histology, evolution, and molecular signatures. Despite occurring at lower frequency, RCRC is often associated with worse diagnosis compared to LCRC. The differences in molecular profiles between RCRC and LCRC also influence the mode of treatment that can be used to specifically target these cancer entities. A better understanding of the cancer cell–CAF interplay and its association with RCRC and LRCR progression will provide better insight into potential translational aspects of targeted treatment for CRC.

## 1. Introduction

Cancer is an evolutionary disease. Colorectal cancer (CRC) is one of deadliest cancers. CRC represents a complex multicellular entity, consisting of cancer (epithelial) cells and the tumour microenvironment (TME). A major cellular component of TME is stromal cells, also known as cancer-associated fibroblasts (CAFs) [1,2]. The accumulation of these activated fibroblasts is linked to poor prognosis of CRC and disease recurrence [3]. Tumourigenesis in the colon is characterised by bidirectional interaction between malignant cells and CAFs which promote cancer proliferation, metastasis and stemness [4,5,6]. 

Emerging evidence indicates that the anatomical region from which CRC arises dictates the survival of the patients and cancer recurrence [7]. Right-sided (proximal) and left-sided (distal) colon and rectal cancer differ in their molecular characteristics. Besides their molecular features, these cancers also can be differentiated based on embryological, biological and anatomical properties. Tumour sidedness is vital, especially in metastatic cases, and is currently used as a marker to determine the efficacy of cancer treatment, such as anti-epidermal growth factor (EGFR) therapy [8,9].

Nowell first proposed the idea that cancer is an evolutionary system in 1976 [10]. Genetic mutation drives biological evolutionary process and promotes biodiversity [11]. Genomic alterations that support carcinogenesis contribute to the co-evolution of the adjacent stroma, including fibroblasts of the TME [12]. The cancer genome is highly heterogeneous. The heterogeneity in the cancer niche can be seen across different type of tumours, between cases in a type of tumour, and even within cancer of an individual. This variation is a result of dynamic twin evolutionary forces on tumour generation and selection [13]. The complex nature of a tumour has imposed a challenge in selecting the best treatment for CRC, which subsequently affects the patient’s prognosis.

As with other various cancers, understanding the evolution process of colorectal carcinogenesis, particularly its relation to the sidedness and interaction with CAFs, will provide insight into cancer heterogeneity and complexity. This will serve as a basis for more targeted therapy for CRC subjects. 

## 2. CRC and Sidedness

### 2.1. CRC

According to GLOBOCAN 2020, CRC is the third most commonly diagnosed malignancy globally [14]. Many CRC cases occur in developed and developing countries. Despite advancements in screening modalities and cancer treatment, the majority of CRC cases are still being diagnosed at the advanced stage. This has contributed to high morbidity and mortality rates of CRC in both men and women. Although cancer is often labelled as a disease for the elderly, epidemiological data have shown an alarming trend of CRC incidence in subjects younger than 50 years old [15]. The risk factors of CRC include older age; hereditary CRC syndromes such as Lynch syndrome, known as hereditary nonpolyposis colon cancer syndrome (HNPCC), and familial adenomatous polyposis (FAP); inflammatory bowel disease (IBD), which includes ulcerative colitis (UC) and Crohn’s disease (CD); as well as obesity, sedentary lifestyle, tobacco smoking, alcohol consumption and unhealthy diet. CRC is also found to be influenced by gender, as higher cases were reported in males compared to females in many countries [16]. Within common cancers, CRC possesses among the highest proportion of familial cases. It is estimated that 30% of CRC cases are represented by inherited forms of this cancer [17]. 

The human colon consists of millions of crypts. Colonic crypt is lined with epithelial cells and they are separated from stromal cells by the extracellular matrix (ECM). In general, the CRC process starts with an aberrant colonic crypt, which evolves into a polyp (neoplastic precursor lesion). Untreated polyps may gradually grow into malignant, cancerous mass over an estimated time of 10–15 years. The cells that give rise to CRC are presumed to be stem cells or stem cell-like cells, also known as cancer stem cells (CSCs). CSCs are formed by the progressive accumulation of genetic and epigenetic changes that lead to inactivation of tumour suppressor genes and activation of oncogenes. CSCs are located at the base of the colonic crypt and drive tumour initiation and development [18,19]. This presents tremendous potential in targeting the pathways implicated in the evolution of CSCs as part of the treatment avenues and preventive measures for CRC [20,21]. 

Based on the unique genes and signalling pathways involved, CRCs are classified into four molecular subgroups, known as consensus molecular subtypes (CMS): CMS1 (MSI immune), CMS2 (canonical), CMS3 (metabolic) and CMS4 (mesenchymal). CMS1 and CMS3 are associated with tumours in the right colon, whereas CMS2 and CMS4 are implicated in cancer in the left colon [22]. CMS classification is actively investigated as prognostic and predictive markers of CRC through various clinical trials. Tumour sidedness and mutation status, such as in RAS and RAF genes, are applied clinically to design systemic treatments [16]. CMS4 is associated with a more aggressive form of CRC, and patients presenting with this subtype exhibit worse prognosis compared to other subgroups [23].

Regarding treatment, there are several conventional measures that can be taken for CRC management, including endoscopic resection, surgery and chemoradiotherapy. Immunotherapy and targeted therapy, which are designed based on tumour molecular properties, started to attract much interest in the past few decades [16]. These medical interventions are predicted to be the treatment of choice due to their effectiveness in specifically eliminating cancer cells, thus significantly improving a patient’s survival. Understanding the molecular profile of CRC would be extremely beneficial for its treatment. For example, *BRAF-V600E* mutant CRC is associated with aggressive tumours and a lack of response to systemic therapy, thus leading to poor prognosis [24]. Different treatment regimens involving triplet chemotherapy with bevacizumab and combinatorial therapy (BRAF inhibitors and anti-EGFR antibody coupled with chemotherapy or MEK inhibitors) may be suggested to improve outcomes, as shown by randomised clinical trials [25,26,27].

The dynamic and heterogeneous nature of CRC is also clearly demonstrated by the spatial and temporal evolution of this cancer [28]. The continuous progression of CRC has led to many complications in managing this disease clinically.

### 2.2. Right-Sided versus Left-Sided CRC

Apart from the molecular classification, CRC can be further divided into right- and left-sided colon tumours. There is no clear distinction on the division between CRC of the left and right side of the colon. The common definition used for right-sided CRC (RCRC) is the cancer proximal to the splenic flexure, and left-sided CRC (LCRC) refers to the cancer at or distal to the splenic flexure [29]. This cut-off point is usually applied, as roughly the distal one-third embryologically arises from the hindgut, whereas two-thirds of transverse colon originate from the midgut. Vascular supply also is used to define the embryologic origin where the superior mesenteric arteries supply the midgut, and the hindgut vascular supply is associated with inferior mesenteric arteries supply [30].

RCRC and LCRC differ significantly in their evolutionary mechanism, progression and influence on treatment outcome. RCRC is usually associated with worse prognosis even in the initial stages of the cancer. This cancer also presents more advanced N stages, greater tumour size, poorly differentiated tumours and higher probability of lymphovascular invasion in comparison to LCRC [31]. A number of randomised controlled trials corroborate the report on the predictive effect of tumours. Additionally, the findings support the fact that lower survival (overall survival—OS, progression-free survival—PFS and objective response rate—ORR) was found in RCRC compared to LCRC with RAS wild-type metastatic CRC [32]. Multi-omics analysis also revealed more prevalent pathway crosstalk in RCRC than LCRC, including an RCRC-specific PI3K pathway, which is commonly linked to the RAS and P53 pathways. RCRC also exhibits hypermethylation in comparison to LCRC. This study also identified various differentially expressed genes (*n* = 253) and differentially expressed miRNAs (*n* = 16) between LCRC and RCRC. A gene of interest, prostate cancer susceptibility candidate 1 (*PRAC1*), which is often associated with hypermethylation, represents the most downregulated gene in RCRC. These data clarify the notion of more aggressive phenotypes in RCRC and heterogeneity within the location-based subclassification of CRC [33]. However, there were conflicting studies on the prognostic indication of primary tumour location, CRC stage and severity of the disease. Warschkow et al. (2016) reported that patients with localised RCRC (particularly stage I and II) present with better prognosis than LCRC [34]. Moritani et al. (2014) found no significant differences in 5-year postoperative disease-free survival (DFS) rates between patients with RCRC and LCRC [35]. These contrasting reports render CRC sidedness and its correlation with prognosis as still much-debated topics. 

Current prognostic and predictive biomarkers for CRC include mutations in RAS family (KRAS, NRAS, HRAS) and BRAF (V600E) as well as microsatellite instability (MSI) status [36,37,38,39]. MSI status is considered as the hallmark status to investigate adjuvant therapy for CRC [40]. The majority of RCRC tumours are signified by high microsatellite instability (MSI-H) [41]. The respective biomarkers and differences between RCRC and LCRC are mapped in Figure 1 [16,30,42].

## 3. Tumour Microenvironment of CRC

### 3.1. CAFs

Activated fibroblasts found predominantly in the vicinity of solid tumour mass, termed as CAFs, are reported to have significant roles in CRC progression [4,5,43]. CAFs, referred to as activated myofibroblasts, are a major cell type in TME. They are described as larger-shaped plump-spindle cells with prominent indented nuclei and elongated endoplasmic reticulum, and a Golgi complex distinct from that of normal fibroblasts [44,45]. In early characterisation of CAFs, lineage exclusion is typically applied, where CAFs are identified as cells that are negatively selected for endothelial, epithelial and leukocyte markers and positive expression of mesenchymal markers such as vimentin [2,46]. However, it is noted that these approaches are deemed less specific. The generation of CAFs from normal resting fibroblasts has been cited as one of the origins of CAFs, which describe tumour desmoplasia where myofibroblasts differentiate and produce collagen matrix, resulting in intratumoural fibrosis [47]. Other origins include epithelial cells via epithelial–mesenchymal transition (EMT) and endothelial cells through endothelial–mesenchymal transition [48]. The expansion of fibroblasts is seen in the early tumour stage where CAFs act as “tumour suppressors”, producing gap junctions which subsequently turn CAFs into “tumour promoters”, as activated by tumour-secreted factors such as PDGF, FAP, interleukin-4, interleukin-6 and prostaglandin E (PGE) [49,50]. Still, the origin of CAFs remains ill-defined, shedding light on the heterogeneity and complexity of CAFs [51,52]. 

CAFs can be identified using an array of biomarkers, including the fibroblast activation protein (FAP), α-smooth muscle actin (α-SMA), fibroblast-specific protein 1 (FSP-1) and platelet-derived growth factor receptor-β (PDGFR-β) [53]. These are widely used markers for CAFs, especially in advanced CRC cases with unfavourable prognosis. The elevated FAP expression, as well as other markers’ expression, including α-SMA and PDGFR-β, are observed in stroma-high compared to stroma-low CRC tissues [54,55]. However, their reliability as specific markers for CAFs is significantly impeded by their heterogeneous expression between CAF subpopulations [53], thus leading to confusing and misleading definitions of these activated fibroblasts. 

Besides the aforementioned classical markers (FAP, α-SMA, FSP-1), there are various emerging biomarkers that can potentially be applied for CAF identification and to predict disease prognosis, as shown in Figure 2 [56,57,58,59,60,61,62,63]. Further work must be performed to clarify the performance of these markers for clinical use. The association between expression of these markers and the anatomical site from which the tumour originates in the colon is yet to be confirmed. Collectively, these potential novel colonic CAF markers are worthy of further study given the unprecedented role of CAFs in carcinogenesis. Additionally, this would help in uncovering the underlying molecular mechanisms that correspond to the aggressive phenotypes in CRC. 

The poor prognosis in CRC patients is linked to the abundance of CAFs rather than the epithelial cancer cells alone, as found in other solid cancers [64,65]. These seminal findings highlight the potential targeting of CAFs as prognostic factors in cancer relapse and untreated CRC patients with poor prognosis. Recent evidence by Herrera et al. (2021) also demonstrated that the CAF gene expression signatures are associated with pro-tumourigenic effects in CRC model [66]. 

### 3.2. Crosstalk between CAFs and Cancer Cells

CAFs have an essential role in promoting cancer evolution through crosstalk with cancer cells, mainly via a vast network of paracrine and autocrine signalling pathways. Table 1 lists major autocrine and paracrine interactions between CAF and cancer cells, facilitated by various secretomes, such as growth factors [3,67,68,69,70,71,72,73,74,75,76,77,78,79,80,81,82,83]. This unique interaction between cellular components in CRC presents exciting translational potential for clinical use in CRC therapy [84]. Interestingly, autocrine signalling loops such as the canonical WNT and TGF-β pathways have also been reported to be involved in CAF activation that contributes to CRC progression [60]. These highly complex signalling pathways are still being actively studied.

## 4. Cancer Evolution

### 4.1. Cancer Evolution and Impact on Tumour Heterogeneity

It is an established notion that cancer pathologically consists of multiple types of cells [85]. Despite many years of cancer research, the intra-tumour heterogeneity (ITH) in tumours has only been recently described at the genomic level [86]. The complex nature of cancer can be seen from the genetic and cellular heterogeneity of tumour tissue. Malignant cells in the cancer niche are not uniform, but usually form different clones that share a similar genotype. Genetic and epigenetic heterogeneity pose a challenge in cancer management, particularly in diagnosis and treatment. An example of this includes a risk of error in sampling, where samples collected may not be representative of the different parts of the tumour [87]. 

Malignant cells derived from a similar tissue source can be stratified into different subpopulations based on their genomic signatures [88]. It is worth noting that mutation phenotype, number and distribution are highly diverse within and across different tumour histologies [89]. Heterogeneity of cancer genomes also results from external forces and leads to the formation of different subclones. TME influences the selection process of cells that are able to survive in often hostile environments, as in the cancer ecosystem [90].

The accumulation of mutations is a signature of the somatic evolutionary process, which promotes tumour proliferation, immune escape and treatment resistance. The dynamic nature of cancer can be studied via evolutionary theory. The evolutionary history can be deduced from tumour molecular profiles. It is an established concept that stepwise somatic mutations and clonal expansion drive the evolutionary process of cancer [91,92]. For the past three decades, significant advancements in cancer research have been observed. Systematic sequencing of cancer genomes, for instance, has uncovered the diversity in the evolutionary process of tumours and unravelled a vast repertoire of cancer genes [93].

Tumour evolution is an intricate process. Gerlinger et al. (2014) proposed that cancer evolution occurs through two mechanisms, namely, microevolution (gradual paths) and macroevolution (major shifts in evolutionary trajectories). Micro- and macroevolutionary events in tumours can be depicted by (a) clonal evolution of cancer cells over time via successive mutation; (b) evolution over time of cancer cells via successive mutation; (c) evolution of cancer cells that undergo chromothripsis (clustered rearrangement of chromosomes); and (d) evolution of cancer cells involving whole genome doubling events [94]. Early molecular work in the 20th century gave rise to the idea of cancer as an independent somatic evolutionary process. Genetic and epigenetic alterations that contribute to tumour growth and expansion have been identified [95,96]. Mutations provide selective advantages for tumour development through direct effective effects. This overrides the secondary effects of mutations that lead to genomic instability and indirectly result in somatic evolution of a tumour. It is postulated that genomic instability is a by-product of direct selective effects, and in certain scenarios has a significant impact on the evolutionary mechanism of cancer [97].

To date, the majority of cancer evolution studies are concerned with genomic changes. Post-translational modification (PTM) of proteins is starting to be highlighted as one of the factors that may influence the evolutionary process of cancer. PTMs lead to the diversification of protein structures and functions beyond what the gene transcripts dictated. PTMs may reversibly or irreversibly change protein properties through biochemical cascades [98]. The most common PTM is glycosylation, involving the polysaccharide chains’ attachment known as “glycans” to proteins [99], which has been implicated in the evolution of multicellular organisms [100]. Considering the complexity of cancer, the changes that occur throughout tumour progression, such as aberrant glycosylation, serve as suitable biomarkers to monitor disease state, staging, prognosis and appropriate treatment. Cancer-associated alterations in glycosylation of proteins include sialylation, alteration in branched-glycan structures and increased expression of “core” fucosylation [101]. Glycans have been demonstrated to play an essential role in the metastasis of cancer, starting from cell detachment from the primary tumour site, intravasation, transportation to different locations and extravasation [102]. 

### 4.2. Tumour–CAF Co-Evolution

CRC evolution is highly dependent on the molecular pathways involved. CRC that arises sporadically might differ from that associated with inherited CRC syndromes and IBD. Boccarelli et al. (2021) reported the positive expression of CAF biomarkers (APOBEC3C, PDGF, IGF, FLI1, TAP2, TRIM2, ANXA1, ENPP2, CDH1, ROCK1, PNP, UBA6) in fibroblasts of UC and CRC in comparison to a healthy group. Heterogeneous phenotypes of CAF versus CRC may be contributed to by these genes and the associated molecular pathways [103]. 

There is also evidence of the effect of oxidative stress in CAFs, characterised by the loss of caveolin-1 (Cav-1) caused by the cancer cells, on inducing genomic instability in adjacent cells through a bystander effect. These “metabolic” and “mutagenic” drivers promote tumour–stroma co-evolution, DNA damage and aneuploidy in malignant cells, which lead to the formation of a more aggressive tumour [104]. Metabolic interaction through oxidative crosstalks between cancer and stroma cells was also supported by other reports. The reactive oxygen species from cancer cells can also promote the trans-differentiation from fibroblast to myofibroblast that will support tumour development and dissemination [105]. 

The evolution process of cancer that leads to metastasis also proves to be an essential aspect to look at, as a lower five-year survival rate (17%) was reported for metastatic colorectal, lung, breast and prostate cancers compared to primary tumours (85%) [106]. The metastatic setting also proves to be challenging as it contributes to high failure rates in cancer treatment (targeted therapy and cytotoxic drugs) [107]. Considering the essential role of CAF in supporting CRC metastasis, it is indeed a study worth pursuing.

Over the years, considerable evidence has been presented regarding the prognostic impact of the two colonic adenocarcinoma subtypes, RCRC and LCRC, which further dictate their biological differences. Considering the complexity of CAFs in CRC and consolidating the findings that highlight the activated fibroblast–cancer cell crosstalk (Table 1), the heterogeneous populations of CAFs may be further diversified according to the distinctions between the two CRC entities (RCRC versus LCRC). 

As stated in the previous section, LCRC is found enriched in CMSs, particularly the CMS4 subtype, which characterises the invasive and metastatic nature of CRC and is mediated predominantly by CAFs [22,23]. The CMS4 subtype, which predominantly consists of mesenchymal cells, correlates with high morbidity and worse prognosis in the RCRC compared to the LCRC [66]. Nevertheless, the correlation between CAFs and both CRC subsections has yet to be fully elucidated, thus warranting future investigations. 

### 4.3. Analyses of CRC Evolution and Heterogeneity

CRC is a heterogeneous and highly complex disease. Genetic factors contribute to variation in the susceptibility risk of human subjects to developing cancer [108,109]. Genomic evolution has contributed to the complex heterogeneity of CRC. High levels of genetic diversity can be caused by a magnification of genetic drift effects, which involve random loss and fixation of genotypes in small populations and the expansion of deleterious mutants [110]. 

Various cell populations with different molecular properties influence the progression of colorectal carcinoma. In brief, cancer evolution modelling can be represented through mathematical modelling, and computational inference includes (a) population dynamics models of tumour initiation and development, (b) phylogenetic methods to illustrate the evolutionary relationship between various subclones in a tumour and (c) probabilistic graphical models to analyse dependencies among mutations. Evolutionary modelling is vital to enhance our understanding of tumour progression and to predict the prognostic value of cancer treatment, particularly targeted therapy [91]. 

Niida et al. (2021) recently reported that using a combination of genomic analysis and mathematical modelling enabled better visualisation of cancer evolution, thus providing a better understanding of carcinogenesis and possible therapy [111]. They proposed that for CRC evolution, driver mutation and subsequent clonal expansion generate multiple clones in early-stage tumours. Subclones from this population that obtain copy number changes have potential to develop into late-stage tumours in which ITH is generated from the neutral mutation accumulation. Moreover, they proposed therapeutic strategies based on the cancer evolution. Upon completion of conventional therapy, treatment-resistant cancer cells will continue to expand and cause relapse. This could be resolved by incorporating adaptive or intermittent therapies to curb cancer relapse via clonal competition. 

To better understand the cancer evolution process, it is paramount to identify the subpopulation of cells present in a tumour niche. Development in single-cell technologies and multi-omics analyses, especially in the past decade, has given scientists insight into specific subgroups of CRC cells, in relation to their anatomical region and evolutionary process. 

Banerjee at al. (2021) investigated the clonal evolution in RCRC, LCRC and rectal cancer (RC) patients using whole-exome sequencing. They established a Darwinian pattern of evolution that gave rise to ITH and showed a more complex and divergent evolution pattern of LCRC and RC than RCRC. Additionally, they found that separate clones give rise to lymph node metastasis (LN) and extranodal tumour deposits (ENTD) [112] Another study by Imperial et al., 2021, corroborates the hypothesis on distinctions between RCRC, LCRC and RC; they employed bioinformatics analysis on the human cancer database (The Cancer Genome Atlas—TCGA) consisting of somatic mutation, mutation hotspots and proteogenomic analyses [113]. Despite the similarity in the detection of *APC*, *TP53* and *KRAS* mutations in all three tumour locations, distinct mutational behaviours are found between RCRC, LCRC and RC that signify their evolutionary trajectories. An interesting finding on synchronous primary right-sided and left-sided colon cancer (sRL-CC) by Hu et al., 2021, showed clear distinctions between the two lesions, including in histological findings, copy number variants (CNVs) and loss of heterozygosity [114]. However, there are few overlapping mutational signatures detected involving single nucleotide variants (SNVs), onco-driver genes and significant mutation genes (SMGs). Opposing trends between RCRC and LCRC also were further supported by Mukund et al. (2020), who reported on the differences in the expression of two genes (*SLC6A4* and *HOXB13*) between RCRC and LCRC [115]. Side specificity is also observed, where more prominent phenotypes were found in RCRC, including post-transcriptional regulation mediated by both RNA-binding proteins and miRNAs. Higher hypomethylation is associated with LCRC, whereas greater hypermethylation of CpG island was found in RCRC. These data from The Cancer Genome Atlas-COAD cohort helped to identify molecular mechanisms in tumourigenesis and the progression of RCRC and LCRC. Collectively, all these studies provide insight into the evolutionary process that gave rise to RCRC and LCRC, and distinctions in their molecular profiles.

Recent breakthrough technologies, such as single-cell RNA sequencing (scRNA-Seq), have also tremendously helped in dissecting fibroblast heterogeneity. Investigation of heterogeneous populations of fibroblasts has provided insight into the clinical importance of these cells in driving disease progression, especially cancer [116,117,118]. Buechler et al. (2021) proposed that fibroblast heterogeneity is heavily influenced by tissue type in the steady state and during disease development [119]. They further constructed a fibroblast atlas based on single-cell transcriptomics data and reported on two universal fibroblast transcriptional subtypes across various tissues. Interestingly, this key finding suggested that these cells can serve as a reservoir to give rise to specialised fibroblasts in healthy and diseased tissues or organs, such as activated fibroblasts in cancer. Table 2 summarises the single-cell analyses of CAFs that may have resulted from the evolutionary process in both human and animal models [120,121,122,123,124,125,126]. These reports clearly exhibit the complexity of CAF populations that are represented by different molecular signatures. However, limited information was reported on the relationship between CRC sidedness and CAF properties. 

## 5. Discussion

Cancer is a result of an evolutionary process. Current therapeutic strategies against cancer not only aim to focus on individual oncogenes as a target but also the evolving nature of tumours [127]. Despite massive advancements in oncology, there is still a poor understanding of CRC evolution, which contributes to their heterogeneity. The need to identify the changes in DNA and RNA of CRC is of paramount importance in clinical research. 

Cancer progression involves interactions between different cell types and TME. As stated in the previous section, numerous reports have demonstrated the interplay between CAFs and cancer cells in driving colorectal carcinogenesis. However, the exact mechanism and molecular pathways that are implicated in the CAF–cancer cell crosstalk until now are still yet to be fully elucidated. 

The evolution of cancer can be unravelled by the discovery of biomarkers. Due to their nature, both cancer cells (including those in RCRC and LCRC) and CAFs are represented by different markers. These markers are utilised not only for diagnosis but also for monitoring of treatment and as prognostic indicators in CRC patients. As cancer progresses, their signature markers also evolve, and this contributes to the richness of tumour niche. To date, although several CAF markers such as α-SMA and FAP have been identified and used widely, more specific and homogeneous biomarkers are yet to be established. However, emerging CAF biomarkers are attracting interest and demonstrate potential to be applied for clinical purposes in the future.

The shifting paradigm over the past two decades, which largely focused on TME components, specifically CAFs and neoplastic cells in synergistically driving tumourigenesis instead of cancer cells alone, has led to many exciting discoveries on targeted therapy against activated fibroblasts in a tumour [128,129,130]. Emerging studies have unveiled the biochemical crosstalk between CRC cells and CAF through CAF-derived factors and signalling pathways, which have become the critical modulators of cancer progression. However, only some have discovered the strong relation to prognostic outcome and ultimately addressed the poor prognosis factors which describe the aggressive phenotypes of CRC found prominently characterised in RCRC. Hence, the prognostic significance of CAF-derived factors in CRC subtypes remains a matter of conjecture. 

In contrary to the much-reported pro-carcinogenic properties of CAF, there are controversial reports that may support the notion of CAF subsets as the suppressor of tumour progression in various organs [131,132,133]. In colitis-associated colonic carcinogenesis mouse model, it was demonstrated that the activation of hedgehog (Hh) signalling in CAFs suppressed tumourigenesis via regulation in BMP pathways and the inhibition of colonic stem cell gene expression [134]. Palangyo et al. (2015) also suggested the role of the IκB kinase/NF-κB (IKK/NF-κB) signalling pathway in the tumour-restraining function of CAF [135]. It is hypothesised that although these fibroblasts start off as tumour-suppressors, as the cancer evolves and influences TME, the stromal cells may acquire pro-tumourigenic properties. Nevertheless, the TME–cancer interplay that contributes to the evolution of tumour serves as an important target for therapy. Targeted therapy may be designed to target certain autocrine and paracrine signalling pathways involved in the bidirectional communication between malignant cells and CAFs, which can potentially abolish CRC tumour progression. CAF-targeted therapy can be used in combination with other drugs targeting tumour cells [136]. This will improve treatment efficacy and patient survival. Another mode of treatment that may be used in CRC is gene therapy to correct defect genes such as *TP53* and *KRAS* and thus control the tumour growth and metastasis [137]. However, little is known about this aspect of treatment modality and its effect on CAF.

The complexity of the two entities of CRC (RCRC and LCRC) and their evolution process are still matters of debate among medical communities. The variation in molecular mechanisms demonstrated by Mukund et al. (2020) highlights the distinctions in colorectal carcinogenesis based on their different anatomical regions (RCRC and LCRC) [115]. It is proposed that specific molecular signatures of RCRC and LCRC will serve as the basis for prospective research to determine drug efficacy and for future translational purposes. Although the genomic signatures of RCRC and LCRC are described in great detail, the relations between these two CRC entities and their TME components, especially CAFs, are not thoroughly investigated.

There is also consideration that needs to be taken on the variation in microbiota composition in the right and left side of the colon which contributes to the differences in cancer progression and the evolution of these two entities. Microbiota variation may be influenced by diet and living conditions of a subject. Microorganisms in LRCR are reported to drive carcinogenesis, whereas those in RCRC are found to be less invasive in nature [138]. In contrast, a report from Phipps et al. (2021) shows distinctive bacterial populations between the right and left colon, although a more consistent microbiome was detected in the presence of colonic tumours [139]. As stated in the previous section, diet proved to be essential in shaping the gut microorganisms, which could subsequently influence CRC progression. The gut microbiome, which differs according to diet and eating habits, is closely associated with geographical location and varies between different ethnicities. This can be associated with the variations in dietary intake between populations [140,141,142]. For example, subjects living on the Mediterranean diet are reported to have a lower risk of developing CRC [143]. A direct link between microorganism composition, diet, ethnicity and sidedness of CRC has yet to be fully established. 

To date, most of the cancer evolution processes have been investigated at the genomic level instead of the proteomic level. Cellular function in the TME has been proposed to be the functional adaptation of protein. Genetic and epigenetic alterations often lead to changes in functional characteristics of regulatory proteins, which subsequently promote the survival and proliferation of neoplastic cells. Pro-carcinogenic properties of cancer cells are selected and influenced by TME by Darwinian natural selection. Further in-depth studies of post-transcriptional events and protein–protein interaction would provide essential information to understanding tumour variation and its clinical behaviour [113]. 

## 6. Future Perspective

Despite years of research, CRC sidedness is still a much debated topic, and its relations to TME are yet to be fully explored. More extensive work ranging from the identification of specific markers of LCRC and RCRC to proteomic profiling, used in concordance with genomic data, would shed light on CRC heterogeneity in the future. Comprehensive profiling at the multi-omics level would serve as a solid foundation for better cancer therapy targeting the cancer evolution process in both RCRC and LCRC and their TME counterparts, specifically CAFs.

## 7. Conclusions

CRC progression occurs through the evolution from normal colon to the formation of polyps and cancerous growth. This dynamic process signifies the development of cancer on the right (RCRC) and left side of the colon (LCRC). Cancer cell–CAF bidirectional communication has been reported to drive colorectal carcinogenesis. Despite the concrete evidence on the role of CAFs in CRC progression, there is limited information on the impact of these activated fibroblasts on the onset and progression of RCRC and LCRC. Variations in molecular signatures of RCRC and LCRC are proposed to influence CAF properties differently, and vice versa. This phenomenon subsequently may lead to differential mechanisms in CRC evolution. The complexity of cancer evolution and crosstalk between various components in the tumour niche contributes to CRC heterogeneity, which hinders the effective management of this cancer. Over the past decade, single-cell technologies have helped tremendously in investigating individual subgroups of cells in the cancer niche, including CAFs and neoplastic cells. These discoveries provide a greater understanding of the cancer niche and the evolution process that gives rise to CRC. Better insight on the crosstalk between cancer cells and CAFs in both RCRC and LCRC will help in designing more targeted therapies for CRC patients in the future. 

## Figures and Tables

**Figure 1 biology-11-01014-f001:**
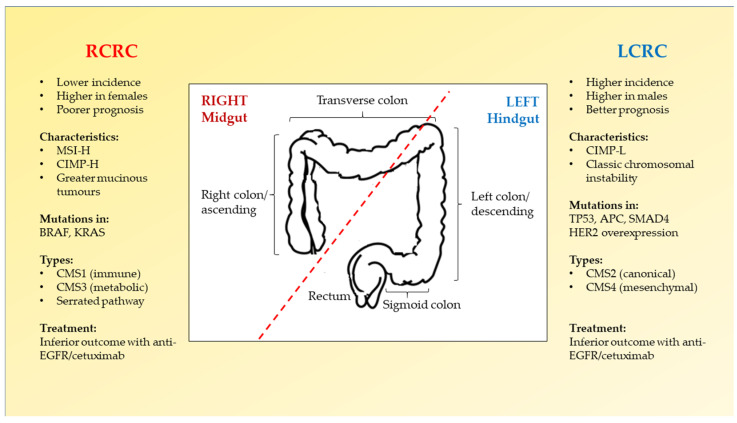
Right-sided CRC (RCRC) versus left-sided CRC (LCRC). RCRC and LCRC can be differentiated according to the anatomy (indicated by the red dotted line), their prevalence, prognostic value and molecular signatures. These factors determine the most suitable treatment for CRC patients to improve their survival.

**Figure 2 biology-11-01014-f002:**
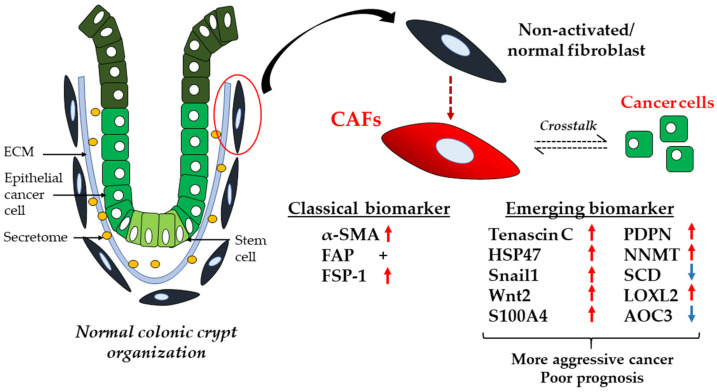
Conventional and emerging CAF markers. CAF transdifferentiation from normal fibroblast (represented by the red circle) and activation are influenced by crosstalk with epithelial cancer cells in the colon. CAFs can be represented by various markers. Classical markers have been applied extensively for CAF characterisation despite their heterogeneous expression. Emerging CAF markers can be potentially used in combination with conventional markers to further dissect the molecular properties of CAFs and to determine prognosis of patients. HSP47: heat shock protein 47; Snail1: Snail family transcriptional repressor 1; Wnt2: Wnt family member 2; S100A4: S100 calcium-binding protein A4; PDPN: podoplanin; NNMT: nicotinamide N-methyltransferase; SCD: stearoyl-CoA desaturase; LOXL2: lysyl oxidase-like 2; AOC3: amine oxidase copper-containing 3; +: positive expression; blue downwards arrow: downregulation of expression; red upwards arrow: upregulation of expression.

**Table 1 biology-11-01014-t001:** Paracrine and autocrine interactions between CAFs and CRC cells in relation to poor prognosis CRC subtype.

Secretome/Mediator		Expression	Influence on Carcinogenesis	Ref.
Chemokine	CCL2; CCL8	Up	Secreted CCL2 and CCL8 from CAFs induce proliferation and invasion of CRC cells	[67]
	CXCL14	Up	Stimulates CAF pro-tumourigenic activity via autocrine effects on CAFs and paracrine signalling on neoplastic cells, leading to higher cancer cell proliferation	[68]
	IL-6/IL-11	Up	Induce tumour proliferation and CAF formation	[69]
			STAT3 activation facilitated by IL-6/IL-11 in CAFs drives CRC progression and is associated with poor prognosis	[70]
			Intrinsic STAT3 activity in CAFs induces the release of IL-6, TGF-β and VEGF by CRC cells and promotes carcinogenesis, immune suppression and metastasis	[71]
	CXCR4/CXCL12	Up	CXCR4/TGF-β1 axis supports the differentiation from HSCs into CAFs and promotes metastasis	[72]
Growth factor	TGF-β	Up	TGF-β activity on CAFs promotes colonisation of CRC cells. TGF-β-stimulated CAFs secrete IL-11, which induces STAT3 signalling that supports cancer metastasis	[73]
			Decreases T-cell activity, leading to cancer immune evasion	[74]
			Presence of upstream transcription factors, SMADs, which predict the failure of immune checkpoint (PD-1) blockade	[75]
			Secreted by CRC cells, interacts with CAF-derived exosome miR-17-5p, resulting in tumour invasion and metastasis	[76]
	IGF-1/IGF-1R	Up	IGFBP7 (TGF-β-target gene) promotes cancer cell proliferation through tumour-stroma paracrine signalling	[77]
			IGF-1 and STAT3 drive CRC progression through cell autonomous and pro-tumourigenic activity of CAFs	[78]
	Wnt/β-catenin	Up	Induce tumour invasion and metastasis	[79]
			CAF-derived WNT2 induces angiogenesis and promotes carcinogenesis	[80]
MicroRNA	miR-135b-5p	Up	Upregulation of miR-135b-5p by CAF-derived exosomes to support CRC cell growth and angiogenesis via TXNIP inhibition	[81]
ECM components	ADAMs	Up	ADAMs expressed by CAFs drive tumour invasion and metastasis	[82]
	TIMP-1		High expression of TIMP-1 stimulates stromal cells growth and activation of ERK1/2 kinase	[83]

ADAMs: disintegrin and metalloproteinases; CCL2/8: chemokine (C-C motif) ligand 2/8; CXCL14: chemokine (C-X-C motif) ligand 14; IL-6/IL-11: interleukin 6/interleukin 11; TGF-β: transforming growth factor-beta; IGF: insulin growth factor; miR-135b-5p: microRNA135b-5p; TXNIP: thioredoxin-interacting protein; VEGF: vascular endothelial growth factor.

**Table 2 biology-11-01014-t002:** Single-cell analysis on fibroblastic cells of CRC.

Purpose	Analysis	Model/Study Design	Finding	Ref.
Studying CRC cellular heterogeneity	scRNA-Seq	Human model	Two distinct subtypes of CAFs (CAF-A and CAF-B) were identified. CAF-B cells showed expression of cytoskeletal genes and other associated markers of activated myofibroblasts, whereas expression of ECM-related genes was found in CAF-A.	[120]
Studying genomic changes of CRC stromal cells	Single-cell multi-omics sequencing	Human model	Higher proportions of aneuploid fibroblasts in tumours compared to those in normal tissues, with significant clonal expansion of fibroblasts with an extra copy of chromosome 7.	[121]
Single-cell analysis of colon biopsy	Droplet-based scRNA-Seq, SMART-Seq2 on colonic spheroids	Human model—normal and UC patients	Using clustering, 51 cell subsets were identified (epithelial: 15; fibroblast: 8; endothelial: 4; glial: 1; myeloid: 7; B: 4; T: 10 (*CD4*^+^ T_conv_, T_regs_, *CD8*^+^, and γδ); innate lymphoid cell (ILC): 1; NK cell: 1. The inflammatory fibroblast (IAF) subset expresses markers of CAFs unique to UC, suggesting an IAF expansion of CRC. IAFs are composed of *WNT2B*^+^ and *WNT5B*^+^ subsets.	[122]
Single-cell transcriptional profiles study	SmartSeq2	Animal (murine) model—comparison between fibroblasts and vascular cells in muscular organs	Subpopulation of fibroblast cells (*Tnc*^+^ *Cd34*^−^) which are localised at the surface epithelium whereas *Tnc*^−^ *Cd34*^+^ fibroblasts were found deeper down in the lamina propria and in the muscularis mucosa. Differential expression in BMP and WNT signalling pathways was also reported between the two populations.	[123]
Prediction of prognosis and therapeutic responses in CRC	GEO single-cell transcriptome, qPCR analyses	Bio-informatics analysis	Established the correlation between greater CAF risk scores with poor prognosis in CRC samples. Those with higher CAF risk scores indicated lower response to immunotherapy, but better sensitivity to conventional chemotherapeutics.	[83]
Classification of tumour cells and clinical stratification	Single-cell resolution transcriptomic analysis	Bio-informatics analysis	Identification of the transcriptional signature of specific subtypes of colorectal CAF (CAF-S1 and CAF-S4) that significantly indicate stratification of a patient’s survival. Two CAF-S1 subpopulations, *ecm-myCAF* and *TGFß-myCAF*, are linked to primary resistance to immunotherapies.	[124]
Association between presence of IL-11-expressing fibroblasts and CRC prognosis	Transcriptome analysis on human cancer database	Bio-informatics analysis	Expression of fibroblast markers and genes implicated in cell growth and repair in IL-11^+^ cells. Expression of genes enriched in IL-11^+^ fibroblasts is increased in colorectal tumours and associated with lower recurrence-free survival.	[125]
Dissecting ITH of CRC	Single-cell exome and transcriptome sequencing	Animal (mouse) model and metastatic human CRC model	Demonstrated the dynamics of ITH of CRC. The emergence of transcriptional subpopulations which lead to increased ITH may be vital for adaptation to drastic changes in the microenvironment when malignant cells have gained sufficient genetic alterations at the advanced stage of tumourigenesis.	[126]

GEO: Gene Expression Omnibus; IAF: inflammation-associated fibroblast; IL: Interleukin; qPCR: quantitative real-time polymerase chain reaction.

## Data Availability

Not applicable.

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
