# Peer review of "Dynamic Co-Evolution of Cancer Cells and Cancer-Associated Fibroblasts: Role in Right- and Left-Sided Colon Cancer Progression and Its Clinical Relevance"

_biology, 2022, doi:10.3390/biology11071014_

Round 1

Reviewer 1 Report

The Biology journal targeted for publication of the article has an effective value and matches the scope of this study.
ï‚· When the entire article is evaluated, colorectal cancer is explained in detail and the effectiveness of the tumor microenvironment is presented in detail. The images have descriptively covered the link in the text.

Author Response

Dear Reviewer, Please see the attachment. Thank you for the kind feedback and comments

Reviewer 2 Report

The manuscript entitled "Dynamic co-evolution of cancer cells and cancer-associated fibroblasts: Role in right- and left-sided colorectal carcinogenesis and its clinical relevance" by Sahira Syamimi Ahmad Zawawi  and Marahaini Musa. It will be improved if the followings are undertaken.

- The article title is a bit confusing as it is unclear what is the meaning of right- and left-sided colorectal carcinogenesis. I suggest you to improve the meaning of the expression.

- Throughout the manuscript, it is overwhelmed with mostly repeated colon cartoons all over the place, it would be better to design other and useful figures to present the content more vividly.

- Would it be the reason of different microorganisms situated in the left and right colon which governs the difference in colon cancer progression and characteristics? Please elaborate more on this aspect.

- Would there be difference in colon cancer susceptibility in people of  different ethnicity (i.e. East Versus West world, and the rest of the humans in this world, male versus female), as the diets are different. Please provide these and elaborate more in the revision.

-The way of the data presentation in the tables is unclear, you should redesign and improve them. Also, more useful information should be added in the tables.

- The heading in section 2, in which the authors typed "2. Review" is weird and should be amended. The same apply to all the subheadings, which should be more informative.

- The word tumour and tumor were used, please use either British or American vocabularies.

- A Perspective section should be added in the last part to forecast the research trend of the subject in the next five years.

- The content of figure legends should be elaborated, now is too simple.

- Typos and unfriendly mode of English usage can be found.

Author Response

(The authors gave the same response as above.)

Reviewer 3 Report

The manuscript deals with a topic of wide scientific interest. However, it has some limitations that authors should try to correct. The manuscript should be structured more broadly since it is a very widespread and well-known topic. It is necessary to broaden the discussion, the references, the English language must be revised and improved and made more fluent and it would be useful for the manuscript to have tables or graphics.

Author Response

(The authors gave the same response as above.)

Round 2

Reviewer 2 Report

Most of my concerns are addressed.

Author Response

Dear Sir/Madam

With regards to reviewer's comment stated as following 'Most of my concerns are addressed', here we re-submitting the manuscript. Minor grammatical errors are highlighted by the editor and addressed as suggested.

Thank you

Marahaini
